# Time- and Sex-Dependent Effects of Fingolimod Treatment in a Mouse Model of Alzheimer’s Disease

**DOI:** 10.3390/biom13020331

**Published:** 2023-02-09

**Authors:** Pablo Bascuñana, Mirjam Brackhan, Luisa Möhle, Jingyun Wu, Thomas Brüning, Ivan Eiriz, Baiba Jansone, Jens Pahnke

**Affiliations:** 1Department of Pathology, Section of Neuropathology, Translational Neurodegeneration Research and Neuropathology Lab, University of Oslo and Oslo University Hospital, Sognsvannsveien 20, 0372 Oslo, Norway; 2Pahnke Laboratory (Drug Development and Chemical Biology), Lübeck Institute of Experimental Dermatology (LIED), University of Lübeck and University Medical Center Schleswig-Holstein, Ratzeburger Allee 160, 23538 Lübeck, Germany; 3Department of Pharmacology, Faculty of Medicine, University of Latvia, Jelgavas iela 3, 1004 Rīga, Latvia; 4Department of Neurobiology, The George S. Wise Faculty of Life Sciences, Tel Aviv University, Tel Aviv 6997801, Israel

**Keywords:** FTY720, fingolimod, Gilenya, APPPS1, Alzheimer’s disease, amyloid beta, treatment

## Abstract

Alzheimer’s disease (AD) is the most common cause of dementia. Fingolimod has previously shown beneficial effects in different animal models of AD. However, it has shown contradictory effects when it has been applied at early disease stages. Our objective was to evaluate fingolimod in two different treatment paradigms. To address this aim, we treated male and female APP-transgenic mice for 50 days, starting either before plaque deposition at 50 days of age (early) or at 125 days of age (late). To evaluate the effects, we investigated the neuroinflammatory and glial markers, the Aβ load, and the concentration of the brain-derived neurotrophic factor (BDNF). We found a reduced Aβ load only in male animals in the late treatment paradigm. These animals also showed reduced microglia activation and reduced IL-1β. No other treatment group showed any difference in comparison to the controls. On the other hand, we detected a linear correlation between BDNF and the brain Aβ concentrations. The fingolimod treatment has shown beneficial effects in AD models, but the outcome depends on the neuroinflammatory state at the start of the treatment. Thus, according to our data, a fingolimod treatment would be effective after the onset of the first AD symptoms, mainly affecting the neuroinflammatory reaction to the ongoing Aβ deposition.

## 1. Introduction

Alzheimer’s disease (AD) is the most common cause of dementia and affects over 30 million people worldwide [1]. AD is characterized by the accumulation of different neurotoxic proteins, e.g., amyloid-β (Aβ) in plaques and tau as neurofibrillary tangles, leading to neurodegeneration, neuronal loss, and irreversible cognitive decline [2,3]. The causes responsible for the accumulation of Aβ and tau in the brain remain unclear. Even though no successful therapeutic strategies have been developed, several molecular mechanisms involved in the pathogenesis of AD have been suggested: Aβ overproduction and impaired Aβ clearance, dysregulated tau phosphorylation, altered glutamatergic neurotransmission, as well as astrocyte and microglia activation prior to the clinical onset [4,5,6]. All of these proposed mechanisms can be used to identify new treatment targets for AD as mono- or multi-targeted therapies.

### Fingolimod

Fingolimod (FTY720, Gilenya; Novartis, Basel, Switzerland), a substrate of sphingosine kinases, targets several of the mechanisms mentioned above. The compound binds to sphingosine-1-phosphate (S1P) receptors in its phosphorylated state [7]. The main pharmacologic effect of fingolimod is immunomodulation by lymphocyte sequestration in the lymph nodes by inhibiting the function of S1P_1_ receptors in lymphocyte egression [8,9]. It was first clinically tested to modulate allograft rejection after a kidney transplantation [10]. However, the S1P_1_ receptor is also involved in diverse functions in the central nervous system (CNS), such as neurogenesis, astrocyte activation and proliferation, and the communication between astrocytes, neurons, and the blood–brain barrier (BBB) [11]. Furthermore, fingolimod regulates the biosynthesis of sphingolipids, which play important roles in neurodegenerative diseases [12]. For these reasons, fingolimod has been tested in different animal models of neurodegenerative and neuroinflammatory diseases (reviewed in [13]). In addition, fingolimod has shown promising inhibitory effects on Aβ toxicity and synthesis in vitro. Fingolimod ameliorated Aβ neurotoxicity in neuronal cultures, reducing the neuronal death probably by increasing the brain-derived neurotrophic factor (BDNF) concentration [14,15]. The multiple pathways targeted by fingolimod suggest that this drug may be a promising therapy for AD.

Fingolimod has been tested previously in animal models of AD. Various authors have reported on the beneficial effects of fingolimod treatments in mouse models of AD. More specifically, it decreased the Aβ production and/or deposition, induced neuroprotection, reduced neuroinflammation, reduced the occurrence of psychosis-like behaviour, and improved the cognitive function [16,17,18,19,20,21,22,23,24,25].

However, Takasugi and colleagues reported contradictory effects in an APP transgenic model (A7 model, human APP695 harbouring Swedish (K670N, M671L) and Austrian (T714I) mutations under control of Thy1.2 promoter of unknown genetic background [26]), showing a decrease in Aβ40, but an increase in Aβ42 [16]. This study is the only treatment that started before the manifestation of the first plaques, and therefore, also before onset of neuroinflammation.

In addition, fingolimod has been shown to reverse most of the Aβ-induced changes in the gene expression of 12-month-old APP transgenic mice with London (V642I) mutation (FVB-Tg[Thy1; APP695/Ld2] [27]), but it had no beneficial effect on the 3-month-old APP transgenic mice [28]. Thus, the adequate timing of fingolimod treatments still needs to be elucidated.

Here, we aim to evaluate the effect of commencing a fingolimod treatment at two relevant time points: (i) before or at the very early stages of plaque deposition and (ii) at advanced stages of plaque deposition when neuroinflammatory activation was already present in our animal model of AD.

## 2. Results

To evaluate the impact of the starting point on the efficacy of fingolimod treatment in AD, we investigated alterations in weight, behaviour, Aβ load, neuroinflammatory reaction, and BDNF in a mouse model of AD at two different ages.

### 2.1. Weight Development during Fingolimod Treatment

The fingolimod treatment in the drinking water did not affect the water consumption (data not shown) or body weight evolution of the animals at any of the ages tested (Figure 1). No significant differences were found.

### 2.2. Effects of Fingolimod on Activity or Spatial Orientation

The fingolimod treatment had no alternating effects on the overall daily activity of the mice neither in the light nor at the dark phase (Appendix A—Figure A1). The spatial orientation performance testing did not reveal significant differences (Appendix A—Figure A2).

### 2.3. Effects of Fingolimod on Aβ Load

When the treatment was started at 125 days of age (late treatment), fingolimod reduced the deposition of insoluble Aβ (9.1 ± 2.6 vs. 12.9 ± 3.6 ng/mg brain, *p* = 0.003) and the accumulation of soluble Aβ (5.5 ± 2.6 vs. 7.9 ± 3.9 pg/mg brain, *p* = 0.006) in males. However, no differences were found in the female animals when we measured only insoluble Aβ load (14.8 ± 2.2 vs. 14.1 ± 1.2 pg/mg brain, *p* = 0.47) (Figure 2).

In addition, the early treated female and male mice did not show differences in insoluble Aβ compared to their respective control groups (starting treatment at 50 days of age; 5.6 ± 0.9 and 4.9 ± 0.7 pg/mg brain vs. 5.1 ± 0.9 and 4.9 ± 0.8 pg/mg brain, respectively) (Figure 3).

### 2.4. Effects of Fingolimod on the Reaction of Microglia and Astrocytes

Assessment of microglia immunostaining against Iba1 revealed a decrease in the microglia coverage (total area covered by microglia; 7.12 ± 0.01% vs. 9.26 ± 0.01%; *p* = 0.004) and microglia density (number of cells per 10 mm^2^ area; 2175 ± 350 vs. 2832 ± 395 cells/10 mm^2^; *p* = 0.004; Figure 4A–D) in the late FTY-treated males compared to their VEH-treated controls. However, the fingolimod treatment had no effect on microglial activation in the female animals, which is similar to the unchanged Aβ levels.

Astrocyte activation was measured as the astrocyte density in the cortex, and it was not significantly altered by the fingolimod treatment for any of the groups (Figure 4E–H).

### 2.5. Effects of Fingolimod on Brain Cytokine Levels

Microglial cells are important immune cells in the brain, and they are the cells that appear to be activated in the brain tissue when Aβ deposition starts. Therefore, we wanted to characterize the effect of fingolimod on the cytokine profile in the brain. Here, we found that the pro-inflammatory cytokine IL-1β level was reduced in the late treated males (2.34 ± 0.48 vs. 3.19 ± 0.76 pg/100 mg brain, *p* = 0.004) (Figure 5). No differences were observed in the other cytokines that were measured (IFN-g, IL-2, IL-4, IL-6, IL-10, MCP-1, and TNF-α) or in other experimental groups (females and younger animals).

### 2.6. Effects of Fingolimod on Cerebral BDNF Levels

Previous studies have reported that fingolimod increased the BDNF concentration in the brain [20,29]. Thus, we quantified BDNF in the brain extracts in all of the experimental groups. We found a trend (non-significant) of decreased BDNF levels only in late treated males in comparison to that of their control group (70.1 ± 17.2 vs. 84.3 ± 19.3, *p* = 0.06). The remaining groups showed no significant difference in the BDNF levels (Figure 6). However, the BDNF concentration positively correlated to the Aβ levels in all of the groups (r = 0.785; *p* < 0.001).

## 3. Discussion

The present study shows that fingolimod has reducing effects on the Aβ load in our animal model of AD *only* when it is administered at a later stage of the disease. Fingolimod was reported previously to have beneficial effects on the animal models of AD [16,17,18,19,20,21,22,23,25]. However, two studies showed no effect of this treatment when it was administered at the early stages in contrast to the older animals [28,30], which is similar to our finding. Furthermore, the beneficial effect of our treatment was only apparent in male mice, while female mice showed no effect in any of the tested treatment paradigms. This sex-selective effect was not observed previously in any of the studies published, except for a study on patients, in which women were reported to have higher occurrence of adverse effects, mainly infections [31]. It is common that the treatment effects are different between male and female individuals. Interestingly, a recent review by the Norwegian Health Institute (FHI: Folkehelseinstituttet) of patient studies that were exclusively performed in Norway between 2017 and 2022 found that there is a substantial lack of systematically analysing the sex-related effects: 133 studies reported sex-related analyses and 178 did not [32]. Metabolically, male and female individuals are different also at different ages [33,34], and thus, the treatment response differences should be assumed as normal. In older age, increasing differences in anabolic hormone production and degradation adds to the apparent biochemical sex differences [35]. Not only biochemical differences, but also differences in the function of the immune system and its response towards Aβ, esp. microglia, could be influenced by sex (reviewed in [36]).

Fingolimod’s main pharmacologic effect is immunomodulation by lymphocyte sequestration, thus reducing the numbers of T and B cells in circulation [8]. It has been shown that effect of fingolimod depends on the inflammatory status of the animals [37]. Therefore, we hypothesized that in order to observe a positive treatment effect, the microglia must be activated before starting the treatment. We proposed to evaluate the effect of the immunomodulatory treatment with fingolimod in two groups of APPtg mice: those that were early on in the disease at 50–100 days (i.e., before extensive microglial activation) and those that were late in its progression at 125–175 days (when microglia are highly activated) [38]. Our study confirms that the fingolimod treatment is only effective in reducing both microglia activation and plaque load when it is administered at the late phase of Aβ deposition. Although S1P receptors are involved in different mechanisms in astrocytes [39,40,41,42], we did not find any alterations in the astrocyte activation of the APPtg mice with the fingolimod treatment.

As fingolimod’s main effect is immunomodulation, we also evaluated its effect on the brain cytokine levels in the APPtg mice. We found that the fingolimod treatment reduced the pro-inflammatory cytokine IL-1β level. The reduction of inflammatory markers has been described previously in other studies with AD models, but also, with other neurodegeneration models. This reduction might be related to the decrease in the activated microglia in the late-treated males, as observed in our study. On the other hand, several studies have found an increase in BDNF after the fingolimod treatment [14,20,29,43,44,45,46,47]. Contrarily, we found even a non-significant trend of decreased BDNF concentration in male mice treated with fingolimod in comparison to that of their vehicle-treated control group. Nevertheless, we observed a direct correlation between BDNF and Aβ levels in all of the groups. This increase in BDNF in APP^+^/PS1^+^ mice (Prp-APPswe/Prp-PS1ΔE9 [48]) has previously been reported [49]. Thus, the decrease observed in the males of the treated group might be sole due to the reduction in the Aβ load in these animals, and not because of a direct effect of the fingolimod treatment on the BDNF concentration.

In summary, we have shown that the fingolimod treatment has beneficial effects on an AD model, but its success depends on the neuroinflammatory state at the beginning of the treatment and the subject’s sex. Thus, according to our data and as previously described in [23], fingolimod treatments would be more effective after the onset of the first AD symptoms, mainly affecting the neuroinflammatory reaction towards Aβ deposits.

## 4. Material and Methods

### 4.1. Animal Models and Breeding Schemes

Heterozygous APP transgenic (APPtg) mice (APPPS1–21 [50]) were used in this study. APPPS1-21 mice have a combined APP (Swedish mutations) and PS1 (L166P mutation) transgene under control of the Thy1-promoter, leading primarily to pathological Aβ production in the frontocortical neurons and the first cortical Aβ plaques at 45–50 days of age, which occurs much later also in other brain regions, but to a significant lesser extend (e.g., hippocampus).

The transgenic breeding programs were maintained with male heterozygous, transgene-positive breeders and C57BL/6J transgene-negative females. The animals were housed in the animal care facility of the Department of Comparative Medicine at the Oslo University Hospital (Norway) with a 12 h/12 h light/dark cycle at a temperature of 22 °C with free access to irradiated food and autoclaved, acidified water. All of the experiments were approved by the competent authorities and conducted according to the European Union Directive and regional laws.

### 4.2. Experimental Design—Treatment Paradigms

Animals were distributed into eight experimental groups (see also Table A1):(a)Early vehicle-treated males;(b)Early FTY-treated males;(c)Early vehicle-treated females;(d)Early FTY-treated females;(e)Late vehicle-treated males;(f)Late FTY-treated males;(g)Late vehicle-treated females;(h)Late FTY-treated females.

Fingolimod (FTY720, Sigma-Aldrich, Darmstadt, Germany) was given dissolved in drinking water at a dose of 1 mg/kg bodyweight per day. The animals had a mean uptake between 2.8 mL (late treated males) and 3.5 mL (early treated males) per day from the drinking water. The FTY720 dose in water was corrected according to water consumption in each group. The treatment was administered for 50 days starting at 50 or 125 days of age, respectively, depending on the experimental group (Figure 7). The body weight and water consumption rate of the animals was controlled weekly to monitor potential side effects and to adjust the dose.

### 4.3. Assessment of Activity and Cognitive Performance

#### 4.3.1. Assessment of Activity

The activity of the mice was recorded with infrared sensor-based technology (TSE Systems GmbH, Berlin, Germany) for a period of 48 h (plus 24 h for acclimatisation). Within this period, the mice were housed individually in small standard cages and a typical habitat. The treatment of the mice was performed daily between 9 a.m. and 10 a.m., which was excluded from the activity analysis. The movement of the mice was detected by infrared sensors located on top of the cages. The movements were integrated over 10 min time intervals. Subsequently, the data were analysed using the Phenomaster Software (TSE Systems GmbH, Berlin, Germany) and displayed cumulatively as a function of time (Figure A1).

#### 4.3.2. Assessment of Spatial Orientation Performance

A modified version of our previously published protocol was applied [51,52,53,54]. Briefly, the mice were trained for six days, with four consecutive trials per day and a short recovery period of 60 s between the trials. On day 7, the mice were subjected to a probe trial, where the hidden platform was removed, and visual cue trials, where the platform was clearly marked with a visual cue to detect severe visual impairments. Data were acquired using EthoVision XT (version 15, Noldus Information Technology BV, Wageningen, The Netherlands). The animals were treated by gavage after behavioural testing each day to avoid interference due to potential acute effects.

### 4.4. Tissue Collection and Processing

The mice were sacrificed by ketamine/xylazine overdose (400 mg/kg ketamine, 40 mg/kg xylazine) and transcardially perfused with ice-cold 0.1 M phosphate-buffered saline (PBS). The brains were removed, and the cerebrums were divided into two hemispheres. One hemisphere was kept in 4% paraformaldehyde (PFA) in 0.1 M PBS for immunohistochemical processing. The other hemisphere was snap frozen in liquid nitrogen and stored in −80 °C until further protein extraction.

### 4.5. Immunohistochemistry Labelling and Morphological Quantification

Formalin-fixed hemispheres were embedded in paraffin and sliced into four-micrometer-thick coronal sections using a rotation microtome (HM355S, Leica Biosystems GmbH, Nussloch, Germany), as described previously [38,51,52,53,55,56,57,58,59,60,61]. The sections (bregma−2.0 mm) were stained for microglia (anti-IBA1, 1:1000, FUJIFILM Wako Chemicals Europe GmbH, 019–19741) or astrocytes (anti-GFAP, 1:500, Agilent, USA, Z033401-2) using a BOND-III^®^ automated immunostaining system (Leica Biosystems GmbH, Nussloch, Germany) with a haematoxylin counterstain (provided with the staining kit, Bond Polymer Refine Detection, DS9800). The sections for anti-IBA1 staining were pre-treated with citric acid for 20 min before staining. For anti-GFAP staining, the Bond Enzyme Pre-Treatment Kit (AR9551, Leica Biosystems GmbH, Nussloch, Germany) was applied to the sections for 10 min.

After staining, the tissue sections were digitized at 230 nm resolution using a Pannoramic MIDI II slide scanner (3DHISTECH Ltd., Budapest, Hungary).

A quantitative analysis of the microglia and astrocytes in the cortex of each animal was performed automatically as previously described by us [38,55,56,61] using deep-learning algorithms generated with the DeePathology™ STUDIO (DeePathology Ltd., Ra’anana, Israel). We generated specific algorithms to identify IBA1+ cells or GFAP+ cells. The algorithms were applied for the ROIs (Figure 8A). The cell density (number per ROI) and relative areal coverage (% cell area per total ROI) were calculated for each animal.

### 4.6. Quantification of Aβ

We performed the quantification of the brain Aβ levels using an electrochemiluminescence immunoassay. Thus, we homogenized the brain hemispheres and soluble and insoluble Aβ fractions were extracted in TBS and guanidine buffer, respectively [51]. Immunoassays were performed using the V-PLEX Plus Aβ42 Peptide (4G8) Kit and an MESO QuickPlex SQ120 machine according to the manufacturer’s instructions (K150SLG, Meso Scale Diagnostics LLC, Rockville, MD, USA). The results were normalized to the sample weight. The brain Aβ42 content was calculated as pg/mg brain.

### 4.7. Quantification of BDNF and Cytokines

We performed a quantification of the BDNF and inflammatory cytokine levels in the brain by immunoassay. To achieve this aim, brain homogenates were extracted using lysis buffer (Phosphate buffer saline + 1% Triton X-100 + phosphatase inhibitor + protease inhibitor) at 2 µL/mg brain homogenate. The samples were homogenized for 30 s (SpeedMill PLUS, Analytik Jena GmbH, Jena, Germany) and centrifuged at 16,000 *g* for 15 min at 4 °C. Supernatants were diluted 4-fold using a working solution (Meso Scale Diagnostics LLC, Rockville, MD, USA), and immunoassays were performed using the U-PLEX Kit for BDNF and cytokines (IL-2, IL-4, IL-6; TNFα, IFNγ) and a MESO QuickPlex SQ120 machine following the manufacturer’s instructions.

### 4.8. Statistical Analyses

All of the statistical analyses were performed using GraphPad Prism 9 software (GraphPad Software, CA, USA). We verified the data for Gaussian normal distribution by using the Shapiro–Wilk normality test. The Student’s *t*-test or Mann–Whitney test were performed to determine the significant differences between the two groups. The correlation coefficient was analysed using Pearson’s correlation test. Data are presented as means ± standard deviation (SD). Differences were considered to be statistically significant when *p* < 0.05. The number of subjects (*n*) is reported in the figure legends and is summarized in Appendix B (Table A1). Detailed statistical methods are summarized in Appendix B (Table A2).

## Figures and Tables

**Figure 1 biomolecules-13-00331-f001:**
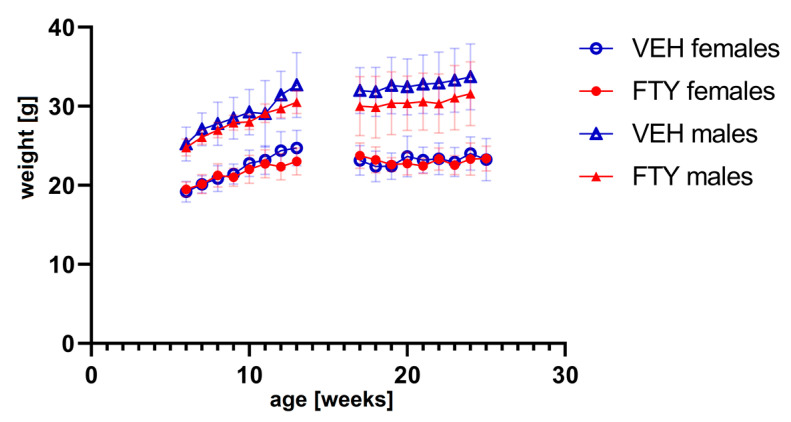
The graph shows the weight progression of early treated (**left**) and late treated animals (**right**) for all experimental groups. FTY—fingolimod (FTY720)-treated animals; VEH—vehicle-treated control animals. Data are presented as mean ± SD; *n* = 8 in all groups, except for the late treatment males (*n* = 13). Statistical analysis was performed using two-way ANOVA (mixed-effects model). No significant differences were found.

**Figure 2 biomolecules-13-00331-f002:**
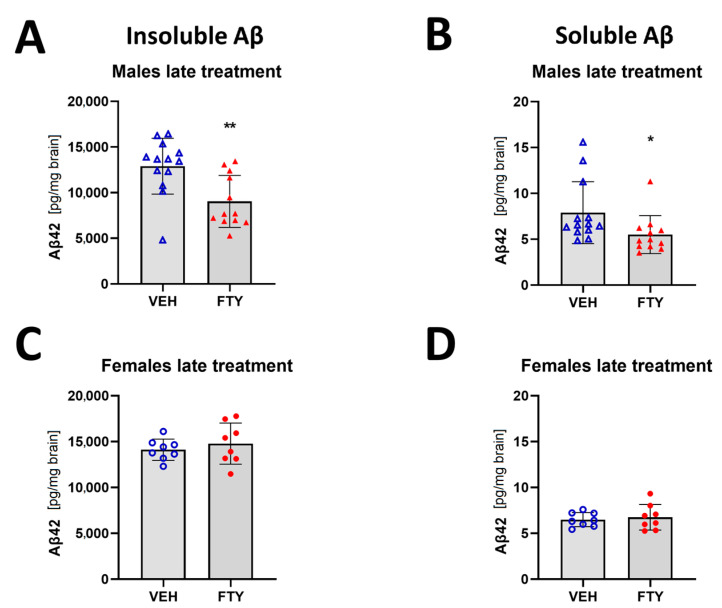
Quantification of insoluble (**A**,**C**) and soluble (**B**,**D**) Aβ42 for late treated males (**A**,**B**) and females (**C**,**D**) APPtg mice. Data are presented as mean ± SD; *n* = 8 (except for males VEH, *n* = 13; FTY, *n* = 12). Significance was calculated using Wilcoxon–Mann–Whitney test, and it is given as *: *p* ≤ 0.05, **: *p* ≤ 0.01.

**Figure 3 biomolecules-13-00331-f003:**
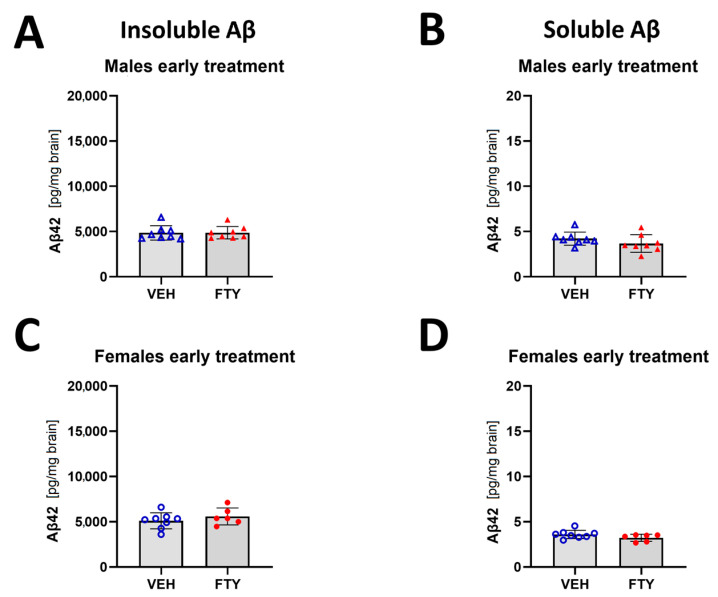
Quantification of insoluble (**A**,**C**) and soluble (**B**,**D**) Aβ42 for early treated males (**A**,**B**) and females (**C**,**D**). Data are presented as mean ± SD; *n* = 8 (except for FTY treatment females, *n* = 6). Significance was calculated using Wilcoxon–Mann–Whitney test.

**Figure 4 biomolecules-13-00331-f004:**
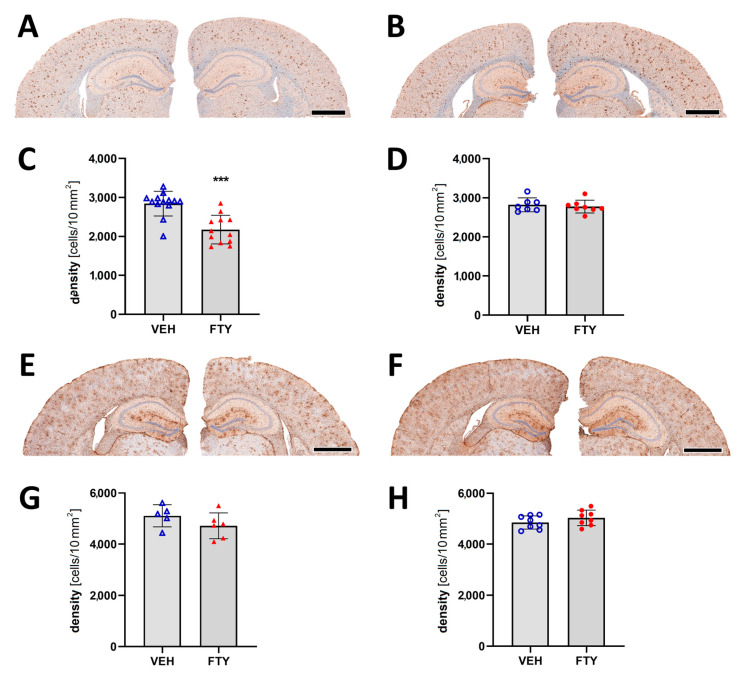
Evaluation of micro- and astrogliosis in mice with late treatment paradigm (at 175 days of age). (**A**,**B**) Representative images of Iba1 immunostaining of male (**A**) and female (**B**) mice. Microglia activation was determined as density of IBA1-positive cells in males (**C**) and females (**D**). (**E**,**F**) Representative images of GFAP immunostaining in males (**E**) and females (**F**). Astrocyte density in males (**G**) and females (**H**). Data are presented as mean ± SD; VEH males *n* = 13 for IBA1, *n* = 5 for GFAP, FTY males *n* = 12 for IBA1, *n* = 6 for GFAP, females *n* = 8. Significance was calculated using Student’s *t*-test, and it is given as ***: *p* ≤ 0.001. Scale bars: 1 mm.

**Figure 5 biomolecules-13-00331-f005:**
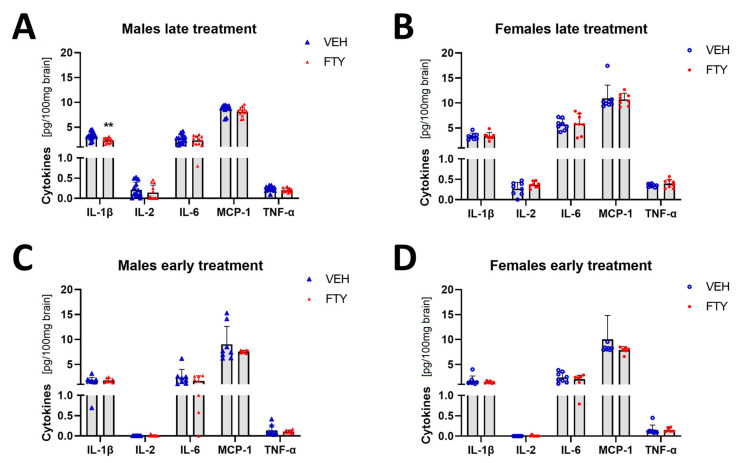
Levels of selected cytokines in brain tissue in late and early treated animals. (**A**,**B**) Quantification of IL-1β, IL-1, IL-6, MCP-1, and TNFα in late treated males (**A**) and females (**B**). (**C**,**D**) Quantification of the cytokines in early treated males (**C**) and females (**D**). Data are presented as mean ± SD; *n* = 8 except late treatment males (VEH *n* = 13; FTY *n* = 12) and early FTY treatment females (*n* = 6). Significance was calculated using Student’s *t*-test, and it is given as **: *p* ≤ 0.01.

**Figure 6 biomolecules-13-00331-f006:**
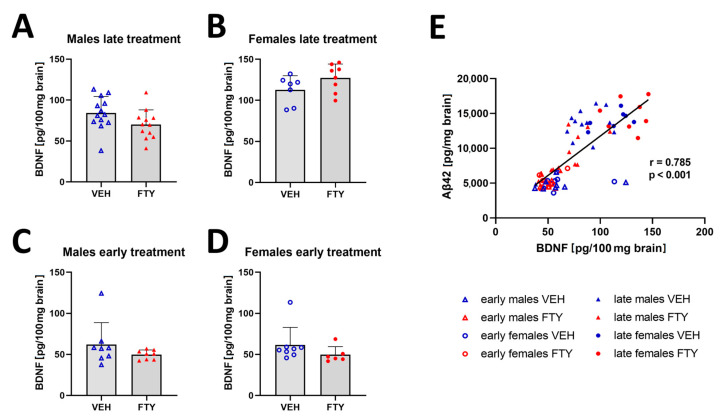
Amount of BDNF in brain tissues. (**A**–**D**) Graphs show the quantification of BDNF in brain tissues of late treated male (**A**) and female (**B**) and early treated male (**C**) and female (**D**) APPtg mice. (**E**) Linear regression analysis between brain BDNF and Aβ concentrations show a positive correlation between both of the proteins. Data are presented as mean ± SD; *n* = 8, except for late treated males VEH, *n* = 13; FTY, *n* = 12) and early treated females (*n* = 6). Significance between experimental groups was calculated using the Wilcoxon–Mann–Whitney test. Correlation coefficient was analysed using Pearson’s correlation test.

**Figure 7 biomolecules-13-00331-f007:**
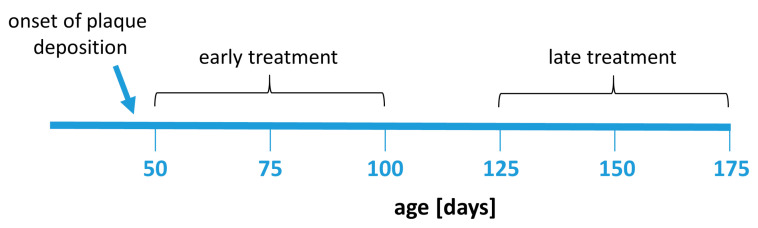
Experimental design. Animals were treated with fingolimod in two treatment paradigms: early treatment (between 50 and 100 days of age) or late treatment (between 125 and 175 days of age). The treatment duration (50 days) was the same in all treatment groups.

**Figure 8 biomolecules-13-00331-f008:**
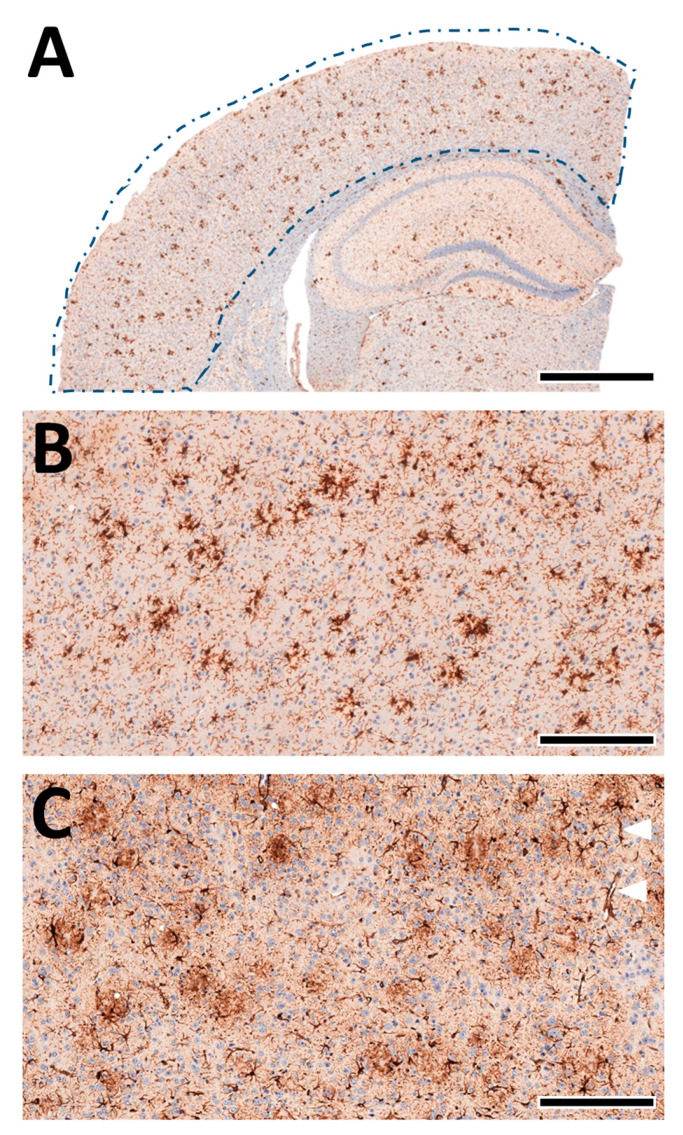
Machine learning-assisted morphological analyses of cortex pathology. (**A**) Schematic presentation of the analysed cortex region (from Figure 4A, left). The brain tissue was automatically detected within the manually marked ROI. (**B**) Staining of IBA1+ cells in the cortex. The microglia nicely accumulate near-insoluble Aβ deposits and delineate amyloid plaques. (**C**) Staining of GFAP+ cells in the cortex. Astrocytes show a more diffuse pattern with localised perivascular pronunciation (white triangles). Scale bars: (**A**) 1 mm, (**B**,**C**) 50 µm.

## Data Availability

Data files and figures can be downloaded from the ABCS1P project at http://www.doi.org/10.17605/OSF.IO/VWQ58 (accessed on 1 February 2023).

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
