# Peer review of "Time- and Sex-Dependent Effects of Fingolimod Treatment in a Mouse Model of Alzheimer’s Disease"

_biomolecules, 2023, doi:10.3390/biom13020331_

Round 1

Reviewer 1 Report

The paper entitled "Time and sex dependent effects of fingolimod treatment in a mouse model of Alzheimer’s disease" partially failed in the highlighting of its most important result. The authors confirmed the effects of the fingolimod in a AD model. Moreover, authors demonstrated that the oral treatment is ineffective in female mice. This is the focal point of this paper. But they poorly commented on the result. They proposed no additional data to sustain this evidence. For example, the authors did not analyse the behavioural scores of mice. They can comment on pharmacokinetic issues or evaluate a change of dose in females. Figure 6 has been analysed in a gender-independent way. Why? If the authors compare BDNF amount in the females (late) and males (late) the difference is clear. The authors can add the supplementary tables to Methods. Moreover, the amount of drinking water or water plus fingolimod /day is lacking.

Author Response

The paper entitled "Time and sex dependent effects of fingolimod treatment in a mouse model of Alzheimer’s disease" partially failed in the highlighting of its most important result.

The authors confirmed the effects of the fingolimod in an AD model. Moreover, authors demonstrated that the oral treatment is ineffective in female mice. This is the focal point of this paper. But they poorly commented on the result. They proposed no additional data to sustain this evidence.

We agree with the reviewer that the different response in both sexes is of interest. However, the focus of our study was to evaluate the effect of the treatment applied in the early and late phases of the disease in our mouse model. It has been already shown in previous studies that neuroinflammatory state could be important for the decision of WHEN to start the treatment with fingolimod.

Thus, we designed the experiment with two groups: one before the appearance of amyloid plaques and a second group after the activation of microglial cells (data from previous studies in our group). We included both sexes in the study, because we think it is important to evaluate potential dimorphisms in therapy. However, in contrast to the time of start of the treatment, there were no data suggesting differences between sexes. In fact, some of the previous studies that are cited in this manuscript are performed in female animals and got beneficial effects when treated with fingolimod. We have searched in the literature and we found no explanation for the difference that we saw. Thus, we believe that it is risky to venture a theory with no hint of the reason behind this difference.

For example, the authors did not analyse the behavioural scores of mice.

We now included behaviour data that we gained from activation and spatial orientation assessments (Appendix A, Figures 9 and 10, also in Methods section). The mice do not show significant behavioural alterations, which is in line with many published experiments where molecular changes do translate into behavioural changes in a reduced way, esp. when models are used, where the neuronal loss is limited even while having severe amyloid deposits. The alterations are very subtle and needed to be assessed by structural assessment of the synaptic distribution.

They can comment on pharmacokinetic issues or evaluate a change of dose in females.

We throughout revised the discussion and included comments on sex-specific topics.

Figure 6 has been analysed in a gender-independent way. Why? If the authors compare BDNF amount in the females (late) and males (late) the difference is clear.

We wanted to evaluate the effect of fingolimod in BDNF concentration as it has been previously shown that fingolimod can increase BDNF concentration. Therefore, we compared each experimental group with its control, including separating both sexes as there might differences between male and females. Actually, BDNF levels are higher in late treatment females compared to late treated males. However, this difference is seen in both, VEH and FTY groups and it is just a reflection of the higher concentration of amyloid in those animals. On the other hand, when comparing group levels of BDNF, we found no differences on BDNF concentration in any of the experimental groups compared to its control (FTY vs VEH). We revised the sentence and updated the graph (Figure 6E) to show each experimental group independently to make this clearer for the readers. Therefore, male and female mice were analysed in one group.

The authors can add the supplementary tables to Methods.

Thanks for this suggestion we now added both tables to the Appendix B section in the manuscript.

Moreover, the amount of drinking water or water plus fingolimod /day is lacking.

Adding the treatment to the drinking water did not change water consumption in any of the groups studied. For example, late-treated males consumed 2.9 ml/animal/day (VEH) and 2.8 ml/animal/day (FTY). Thus, we did not consider this data of value for the manuscript. We have now included additional information.

Reviewer 2 Report

The main objective of the manuscript is to determine the temporal and sex differences in a mouse model of Alzheimer's in response to fingolimod.

The manuscript is generally easy to read, the figures are precise, and the observations are explained.

Unfortunately, due to this simplicity, the authors leave out of the discussion of their results some elements that seem very relevant when drawing appropriate conclusions.

One of the emerging elements is related to the spatiotemporal conversation of the measurements made. Therefore, the authors should address, experimentally or at least in the discussion, the following questions: In their mouse model of AD: Are there differences in Abeta aggregation in different brain areas? Is this affected in mice treated with FTY?

The progression in the appearance of damage markers is very relevant in this study, but the spatial element is transcendental to generate better conclusions from the results presented.

The authors show a clear effect on microglial activation in late-treated FTY animals. How is it explained that this marked effect is not correlated with appreciable differences in cytokine levels?

Another element that is not addressed in the discussion is the differences between different sexes. What is the mechanism that could explain these differences?

Specific

In Figure 4, the histology analysis could be significantly improved by considering the following:

-Mark the area to be quantified

-Add a magnification of astrocytes/microglia of the quantified area

-In addition to density, does the number of glial projections change?

Figure 6E is presented to address the positive correlation between Ab1-42 and BDNF. What happens if you compare the control animals with those treated with FTY?

Author Response

The main objective of the manuscript is to determine the temporal and sex differences in a mouse model of Alzheimer's in response to fingolimod. The manuscript is generally easy to read, the figures are precise, and the observations are explained. Unfortunately, due to this simplicity, the authors leave out of the discussion of their results some elements that seem very relevant when drawing appropriate conclusions.

One of the emerging elements is related to the spatiotemporal conversation of the measurements made. Therefore, the authors should address, experimentally or at least in the discussion, the following questions: In their mouse model of AD: Are there differences in Abeta aggregation in different brain areas? Is this affected in mice treated with FTY?

We included additional information about the APPPS1-21 mouse model and the location of expression of the APPPS1 transgenes in the Material and Methods section (4.1). We also included information about the timely deposition.

The progression in the appearance of damage markers is very relevant in this study, but the spatial element is transcendental to generate better conclusions from the results presented.

The authors show a clear effect on microglial activation in late-treated FTY animals.

How is it explained that this marked effect is not correlated with appreciable differences in cytokine levels?

We performed cytokine immunoassay expecting to see the treatment effect as fingolimod is considered an anti-inflammatory drug. We also expected to see a parallel reduction of cytokines together with the reduction of microglia activation. However, the low concentration of these cytokines in brain tissue was in many cases below our detection limit. Thus, we could only analyse some of the cytokines of interest. We observed a reduction of IL-1β in late-treated animals in parallel with a reduction of amyloid. Other cytokines show a non-significant reduction (TNF-α and MCP-1) which might be significant with improved detection limit. 

Another element that is not addressed in the discussion is the differences between different sexes.

What is the mechanism that could explain these differences?

There are no data in the literature suggesting differences between sexes in fingolimod treatment. In fact, previous studies cited in this manuscript are performed in female animals and got positive effects when treated with fingolimod. We have searched in the literature and we found no mechanism that could explain the differences we found. We theorize that this dimorphism is specific of our animal model, as we have also previously seen higher amounts of amyloid in female mice in this model.

Specific:

In Figure 4, the histology analysis could be significantly improved by considering the following:

-Mark the area to be quantified

We quantified the total cortex using AI, not just parts of it. We revised the manuscript and added the Figure 8A to make the ROI clearer in the Methods section.

-Add a magnification of astrocytes/microglia of the quantified area

              We added two example pictures (Figure 8 B,C)

-In addition to density, does the number of glial projections change?

            We did not analyse separately glial projections.

Figure 6E is presented to address the positive correlation between Ab1-42 and BDNF.

What happens if you compare the control animals with those treated with FTY?

When comparing group levels of BDNF, we found no differences on BDNF concentration in any of the experimental groups compared to its control (FTY vs VEH). When the linear regression was calculated independently for each experimental group, only late-treatment male groups shown a positive correlation between BDNF and amyloid concentration. However, these were the larger groups (n=12 and 13). Thus, that the other groups did not show that correlation might be due to the small number of individual points when dividing the groups.

Round 2

Reviewer 1 Report

Dear authors,

I understand your difficult to produce new data on the female trend. Therefore, I appreciate the revised version, and I evaluate this paper suitable for the publication.